# SyncTREE: Fast Timing Analysis for Integrated Circuit Design through a Physics-informed Tree-based Graph Neural Network[*]

**Yuting Hu**
University at Buffalo
Buffalo, NY, USA
`yhu54@buffalo.edu`

**Jiajie Li**
University at Buffalo
Buffalo, NY, USA
`jli433@buffalo.edu`

**Florian Klemme**
University of Stuttgart
Stuttgart, Germany
`klemme@iti.uni-stuttgart.de`

**Gi-Joon Nam**
IBM Research
Yorktown Heights, NY, USA
`gnam@us.ibm.com`

**Tengfei Ma**
Stony Brook University
Stony Brook, NY, USA
`tengfei.ma@stonybrook.edu`

**Hussam Amrouch**
Technical University of Munich
München, Germany
`amrouch@tum.de`

**Jinjun Xiong**
University at Buffalo
Buffalo, NY, USA
`jinjun@buffalo.edu`

## Abstract

Nowadays integrated circuits (ICs) are underpinning all major information technology innovations including the current trends of artificial intelligence (AI). Modern IC designs often involve analyses of complex phenomena (such as timing, noise, and power etc.) for tens of billions of electronic components, like resistance (R), capacitance (C), transistors and gates, interconnected in various complex structures. Those analyses often need to strike a balance between accuracy and speed as those analyses need to be carried out many times throughout the entire IC design cycles. With the advancement of AI, researchers also start to explore news ways in leveraging AI to improve those analyses. This paper focuses on one of the most important analyses, timing analysis for interconnects. Since IC interconnects can be represented as an RC-tree, a specialized graph as tree, we design a novel tree-based graph neural network, SyncTREE, to speed up the timing analysis by incorporating both the structural and physical properties of electronic circuits. Our major innovations include (1) a two-pass message-passing (bottom-up and top-down) for graph embedding, (2) a tree contrastive loss to guide learning, and (3) a closed formular-based approach to conduct fast timing. Our experiments show that, compared to conventional GNN models, SyncTREE achieves the best timing prediction in terms of both delays and slews, all in reference to the industry golden numerical analyses results on real IC design data.

## 1 Introduction

Electronic design automation (EDA) tools are indispensable for designing today's complex integrated circuits (ICs) that can have multi-billion transistors (switches) and logic gates. EDA tools can be

---

[*]Corresponding authors: Hussam Amrouch (amrouch@tum.de), Jinjun Xiong (jinjun@buffalo.edu)

37th Conference on Neural Information Processing Systems (NeurIPS 2023).

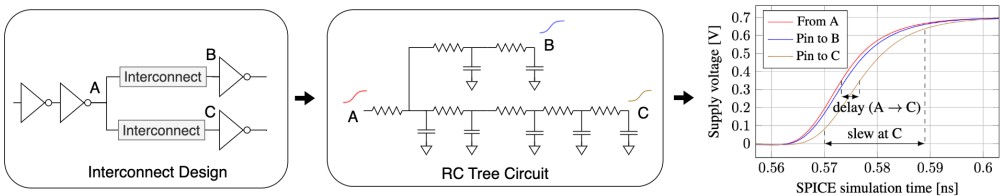

Figure 1: Timing analysis for circuit interconnects.

classified as two categories: the analysis tools that help to evaluate an IC design's quality metrics (e.g., timing performance, signal integrity, and power efficiency), and the optimization tools that use the metrics to guide the various design optimization choices such as logic synthesis, transistor sizing, gate placement, and interconnect layout optimization. Those analyses are often based on physical properties of electronic circuits, such as the Kirchhoff's Current Law (KCL), Kirchhoff Voltage Law (KVL), and electronic components constitutive equations (i.e., the voltage-current characteristics of electronic components such as resistance R, capacitance C, and transistors).

To obtain the most accurate (and golden) analysis results, the industry de factor tool is SPICE ("Simulation Program with Integrated Circuit Emphasis"), which solves a set of differential algebraic equations (DAEs) derived from KCL and KVL via numerical integration methods. Though SPICE simulation is accurate, it is also notoriously slow and not applicable to large-scale ICs' design optimization. Therefore, most EDA analyses tools resort to some mathematical approximation techniques while still follow the physical principles to solve the DAEs in order to strike a balance between accuracy and speed, because those analyses need to be carried out many times throughout the entire cycle of IC design optimization. With the advancement of AI, researchers also start to explore news ways in leveraging AI to improve those traditionally physics-based analysis methods [1].

This paper focuses on one of the most important analyses, timing analysis, for IC interconnects (a.k.a. wires) as shown in Figure 1. For a modern IC design, there are billions of on-chip interconnects that connect billions of transistors and gates, and the total wire length can be as long as hundreds of kilometers. The interconnects help to propagate electronic signals (as represented in a voltage waveform) from a driving gate (the source) to its various downstream receiving gates (the sinks), and the interconnects can be modeled as a distributed RC-tree circuits as shown in the middle of Figure 1. In RC tree, a path is composed of all nodes and edges from the driving source (e.g. A) to a specific leaf node or sink (e.g. B and C). Path delay is defined as the time delay of a voltage waveform propagating through the path measured at the 50% of the waveform's voltage level. At the leaf node, the transition time of a rising voltage waveform from the 10% voltage level to the 90% of voltage level is defined as the waveform's slew (similarly, a falling voltage waveform's slew is defined as the transition time from 90% to 10% voltage levels). Delays capture how fast the voltage waveform propagates from the source to the sinks, and slews reflect how well the propagated voltage waveform looks like at the sinks, that is why both delays and slews metrics are of utmost importance in IC design.

Since tree is a specialized graph, it seems logical to apply graph neural networks (GNNs) [2] to learn a fast timing analysis model from many of known interconnect RC-trees. However, a direct application of modern GNN models will not work well because they do not take into the physical properties of the underlying electronic circuits. For example, as mentioned in prior works [3, 4, 5, 6], most message-passing GNNs cannot retrieve global context dependencies because of the over-smoothing effect. To address this problem, in this paper, we propose a novel tree-based GNN model based on a directed two-pass message-passing mechanism, SyncTREE as shown in Figure 2 to speed up the interconnect RC-tree's timing analysis.

In summary, our major contributions are:

- We formulate a novel GNN model for circuit timing prediction which can provide powerful expressiveness by incorporating global and subbranch dependencies during message passing. To better guide the graph learning process, we further design a new contrastive loss based on Circle loss [7], called Tree Contrastive (TC) loss, to take advantage of the monotone properties of timing delays along an RC path. Importantly, to our best knowledge, this is the first closed-form solution of leveraging GNNs to perform circuit timing analysis.

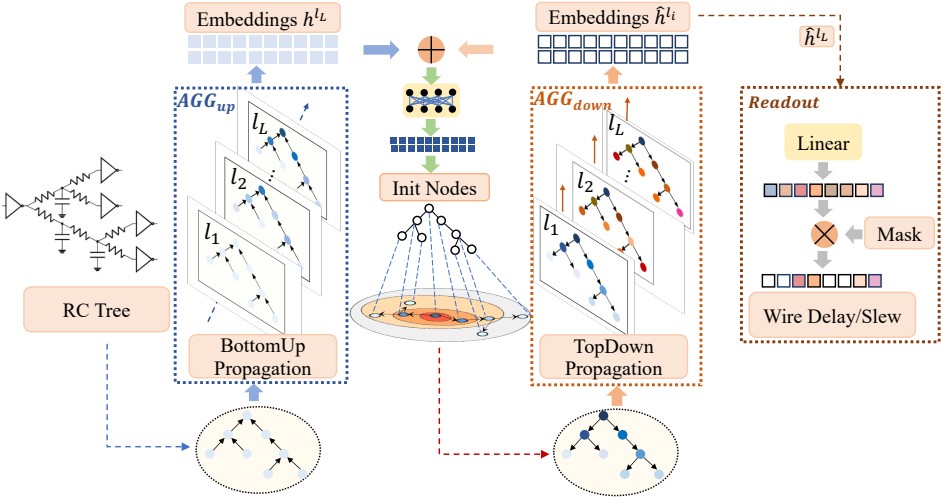

Figure 2: Architecture of our SyncTREE model.

- We create two new interconnect data sets and evaluate our model on them, one synthetically generated and one extracted from the real RISC-V IC designs. According to the experiment results, SyncTREE achieves the best accuracy for both delay and slew prediction when compared with modern GNN models, such DeepGCN[8], GraphTrans[9], and NeuralTree[10]. Furthermore, the experiments validate the advantages of TC loss in guiding our model's learning over Circle loss.

- We analyze the time complexity of SPICE and our model. From the time cost measurement results of the SPICE simulation and our model's inference, it is shown that our model can effectively speed up the timing process with little time increase along with circuit size. Besides, We further evaluate our model's expressiveness by transfer learning experiments on delay and slew prediction tasks. The results show that the representation learned from delay prediction task can effectively transfer to slew prediction task with better performance.

## 2    Related Work

**GNNs in IC Design.**    Recently, AI for EDA has become a cutting-edge research topic. Among learning-based IC design methodologies, GNNs exhibit great potential in enhancing IC design workflows. Some recent researches leverage GNNs (1) to speed up optimization in physical design [11], layout [12], placement[13], and analog design [14]; (2) to improve the prediction analysis of net length [15], timing [16], and routing congestion [17]; and (3) to even produce configuration and guidance in floorplanning [18] and 3D integration [19].

**Other Related GNNs.**    Despite the success of GNNs, most of them are limited to capture long-range context information. Inspired by residual neural networks (ResNets), Li et al. [8] proposed the DeepGCN model to gain a large receptive field with skip connections and dilated convolutions. This allows for the training of very deep GCNs with alleviated gradient vanishing issues. Recently, motivated by the great success of Transformers, Wu et al. [9] proposed the GraphTrans model to learn representations of long-range context. It adds a CLS token to aggregate all pairwise interactions to obtain global graph embedding, and it achieves state-of-the-art results on several graph classification tasks. Although deep GNNs can broaden the receptive fields, it may also lead to node representations that are indistinguishable across the entire graph. Some other approaches are proposed is to enhance node features by incorporating rich information of important subgraph structures. For instance, Neural Tree [10] is a special GNN model that incorporates sub-structural information into feature embeddings by performing message passing on a pre-processed H-tree instead of the original input graph. The H-tree is hierarchically organized from top to down by different subgraphs which are decomposed by the junction-tree algorithm. The benefit of H-tree is that it connects each node to

its sub-graphs which makes the Neural Tree model can learn from substructures. However, it will confuse the global structural information because of its decomposition-based methodology.

**Graph Contrastive Learning.** Being popular in the visual representation learning field, contrastive learning [20] aims to learn discriminative representations for positive and negative samples constructed by the data augmentation process. For graph representation, recent works [21, 22, 23, 24] try to perform contrastive learning to capture rich structure information and make features in agreement with expected transformations in a supervised manner.

## 3 Problem Statement

To obtain the golden results accurately for an electronic circuit, the industrial de facto simulation tool is SPICE [25], through modified nodal analysis (MNA) [26], which builds a system of algebraic differential equations (DAEs). For an arbitrary RC circuit, the DAEs can be constructed by following the physical constraints of KCL, KVL, and the electronic components' branch constitutive equations (i.e., the voltage-current characteristics of resistance R, capacitance C, and voltage sources E), and it can be written as

$$\begin{bmatrix} A_C C A'_C & 0 \\ 0 & 0 \end{bmatrix} \frac{d}{dt} \begin{bmatrix} e \\ i_V \end{bmatrix} + \begin{bmatrix} A_R G A'_R & A_V \\ A'_V & 0 \end{bmatrix} \begin{bmatrix} e \\ i_V \end{bmatrix} = \begin{bmatrix} 0 \\ E \end{bmatrix} \tag{1}$$

where $C$ is the diagonal matrix containing all capacitance; $G$ is the diagonal matrix of all conductance (i.e., the inverse of the resistance $R$); $E$ is the vector of voltage sources' values; $A_C$, $A_R$, and $A_V$ are the incidence matrices built from element types of capacitance $C$, resistance $R$, and voltage source $V$, respectively; $i_V$ is the unknown current vector through voltage sources, and $e$ is the vector of unknown node voltages. The propagation delays and slews of voltage waveforms at all sink nodes can be obtained by solving the above DAEs via numerical integration methods. SPICE usually applies a time-discretization scheme (e.g., the backward Euler method) to solve the DAEs at every time step numerically, leading to particularly intensive computations and long runtime.

**Problem.** Path delays and slews are two critical timing metrics that guide almost every optimization step in an IC design flow, and how to compute them accurately and quickly is of utmost importance. Accurate circuit simulation through SPICE is possible, but it would take very long time. Moreover, as the IC design process is iterative with incremental changes to the interconnect's RC parameters, SPICE simulation cannot take advantage of such incremental updates. Our objective is to design a fast and accurate timing prediction model by leveraging the powerful representation ability of GNNs.

## 4 Physics-informed SyncTREE for Interconnect Timing Analysis

Since electronic circuits have complicated dynamics, it is a big challenge for a GNN model to represent rich structural information at different levels. The key idea of our SyncTREE model is to construct a two-pass message-passing for our tree-based graph representation.

First, we define an undirected graph $\mathcal{G} = (\mathcal{V}, \mathcal{E})$ corresponding to an RC tree extracted from IC interconnects, where each vertex represents a grounded capacitance, and each edge represents the resistance between two vertices. Instead of relying on the undirected graph representation for the RC-tree $\mathcal{G} = (\mathcal{V}, \mathcal{E})$, we derive two directed graph representations, $\mathcal{G}_{bu} = (\mathcal{V}, \mathcal{E}_{bu})$ and $\mathcal{G}_{td} = (\mathcal{V}, \mathcal{E}_{td})$, with $\mathcal{G}_{bu}$ representing the directed bottom-up tree while $\mathcal{G}_{td}$ representing the directed top-down tree. For simplicity, we will use the general graph representation $\mathcal{G} = (\mathcal{V}, \mathcal{E})$ to discuss the common features between $\mathcal{G}_{bu}$ and $\mathcal{G}_{td}$ whenever there is no ambiguity. Otherwise, we will denote them explicitly. Given a graph $\mathcal{G} = (\mathcal{V}, \mathcal{E})$, the feature of each node $i \in \mathcal{V}$ in hidden layer $l$ is represented by vector $h_i^l \in R^{d_l}$. We adopt the general graph kernel from Graph Attention Networks (GAT) [27] as our message-passing backbone. We will leave the exploration of other GNN message-passing mechanisms as our future work.

### 4.1 SyncTREEE Overview

Figure 2 shows the overall model architecture (or algorithm) for SyncTREE. The RC-Tree circuit is represented as two separate directed graphs, $\mathcal{G}_{bu}$ for the directed bottom-up tree and $\mathcal{G}_{td}$ for the directed top-down tree. We first perform GAT like message-passing on the bottom-up graph $\mathcal{G}_{bu}$ by

following the bottom-up directed edges. In doing so, we propagate information from the leaf nodes, through the sub-branches, toward the source nodes. The level of influences will depend on the depth of layers $l_L$ as used in GAT. At the end, every node's embedding is noted as $h_{bu}^{l_L}$. We then perform GAT like message-passing on the top-down graph $\mathcal{G}_{td}$ by following the top-down directed edges. Different from the bottom-up pass, we first copy the hidden feature of $h_{bu}^{l_L}$ from the bottom-up graph $\mathcal{G}_{bu}$ to initialize the corresponding nodes' features in the top-down graph $\mathcal{G}_{td}$. After that, we apply GAT like message-passing top-down on the top-down graph $\mathcal{G}_{td}$ with its own depth of layers $l_L$. At the end, we treat the node representations of the last layer in $\mathcal{G}_{td}$ as the final GNN embeddings and then feed it into the linear layer for readout.

## 4.2 Details of Message Passing in SyncTREE

Inside basic Message-passing layers, we use GAT as the basic block to collect information. Since we treat the node capacitance as the node's attribute and wire resistance as the edge attribute in RC tree graphs, in order to preserve wire resistance information in embeddings, we modify the aggregation mechanism inside GAT to linearly combine node feature $h_j$ and edge feature $e_{ij}$ with normalized attention coefficients $\alpha_{ij}$ as the final output features for every node. The update of the node state after a single hidden layer follows:

$$\alpha_{ij} = \frac{\exp(\text{LeakyReLU}(\mathbf{a}^T[\Theta h_i || \Theta h_j || \Theta_e e_{ij}]))}{\sum_{k \in \mathcal{N}(i) \cup i} \exp(\text{LeakyReLU}(\mathbf{a}^T[\Theta h_i || \Theta h_k || \Theta_e e_{ik}]))} \tag{2}$$

$$h_i^{l+1} = \text{AGG}^l(\{(h_j^l, e_{ij}) | j \in \mathcal{N}(i) \cup i\}) = \sigma \left( \sum_{j \in \mathcal{N}(i) \cup i} \alpha_{ij}(\Theta h_j^l + \Theta_e e_{ij}) \right) \tag{3}$$

where $\Theta$ and $\Theta_e$ are separately applied to every node and every edge as linear transformations, $||$ represents concatenation, $T$ is transpose operation, and $\mathcal{N}(i) \subseteq \mathcal{V}$ is the neighbor node set of $i$.

In SyncTREE, we respectively applied $\text{GAT}_{bu}$ and $\text{GAT}_{td}$ to perform message passing on the bottom-up tree $\mathcal{G}_{bu}$ and the top-down tree $\mathcal{G}_{td}$, respectively.

The feature representation in $\mathcal{G}_{bu}$ is updated by $h_{i,bu}^{l+1} = \text{AGG}_{bu}^l(\{(h_{j,bu}^l, e_{ij,bu}) | j \in \mathcal{N}(i) \cup i\})$. When we start the top-down message-passing, we use the aggregation output of the final layer in $\mathcal{G}_{bu}$ to initialize the node features in $\mathcal{G}_{td}$:

$$h_{i,td}^0 = h_{i,bu}^L \tag{4}$$

For every convolutional layer in $\mathcal{G}_{td}$, the node hidden states will be updated by two parts: node representations updated from the previous layer of $\mathcal{G}_{td}$ itself and the copied final embedding from $\mathcal{G}_{bu}$. It can be expressed as:

$$h_{i,td}^{l+1} = \text{AGG}_{td}^l(\{(h_{j,td}^l, e_{ij,td}) | j \in \mathcal{N}(i) \cup i\}) + h_{i,bu}^L \tag{5}$$

Since $\text{AGG}_{bu}$ and $\text{AGG}_{td}$ are performed on two directed graphs, SyncTREE can effectively avoid over-smoothing and preserve variance among nodes. In our case, we aim at predicting path timing, thus we use the final predictions at leaf nodes (sinks) by applying a mask to calculate the loss function.

## 4.3 Tree Contrastive Loss

It is a well-known fact that the voltage along an RC path is a monotonic function of distance [28]. For a target node, in its RC path to the root node, its timing would be similar to its nearest node and have bigger differences with further nodes. Inspired by contrastive learning, we introduce tree contrastive (TContrast) loss to enhance node embedding quality by optimizing the pair similarity of node representations. The idea behind contrastive learning is to pull positive samples close to the target node and push the negative samples away.

At every batch during training, we randomly sample a set of target leaf nodes with size $B$. For any target leaf node $i \in B$, we get a node embedding set $\mathcal{S}_i$ by picking the final representations $h_{i,td}^L$ of positive and negative nodes from $\mathcal{V}$. To determine positive and negative samples of a target leaf node, we set a $hops$ hyperparameter. On the RC path from the target node to the source node, we treat the nodes within the $hops$ to the target as positive samples and the others as negative samples.

It should contain $M$ positive samples and $N$ negative samples, which means that there will be $M$ pair of within-class similarity $s_p$ and $N$ pairs of between-class similarities $s_n$. We apply Euclidean distance to measure similarity as follows:

$$s_{ij} = \frac{1}{1 + \alpha * \left\| h_{i,td}^L - h_{j,td}^L \right\|_2}, \ h_{j,td}^L \in \mathcal{S}_i \tag{6}$$

where $\alpha$ is used to avoid $s_{ij}$ being too small. For pair similarity optimization, it is natural to maximize within-class similarity $s_p$ and minimize between-class similarity $s_n$. A novel optimization manner of Circle loss is to reduce $(\alpha_n s_n - \alpha_p s_p)$, where the $\alpha_n$ and $\alpha_p$ are independent weight factors. The intuition behind it is different similarity scores should have different penalty strengths. For the Circle loss, it aims to optimize $s_p \rightarrow 1$ and $s_n \rightarrow 0$, and use $m$ the radius of the circle decision boundary [7]. However, it's unfair for the Circle loss to use the same relaxation strength $m$ to $s_p$ and $s_n$. In TContrast loss, we design a more flexible decision boundary based on the priori path resistance of samples. For each $(s_p^k, s_n^q)$ pair, we define a coefficient $r_d^{k,q}$ of path resistances between positive node $k$ to target $i$ and negative node $q$ to target $i$ as follows:

$$r_d^{k,q} = \frac{\sum_{j=i}^k R_j}{\sum_{j=i}^q R_j} \tag{7}$$

where the numerator/denominator is the sum of resistance in RC path $i$ to $k/q$. Instead of only using similarity score to amplify the gradient to $s_p$ and $s_n$, we separately set distance-adaptive scaling factors based on path resistance for $\alpha_n$ and $\alpha_p$ by:

$$\alpha_p^{k,q} = e^{r_d^{k,q}}[(1 + m - s_p^k)]_+, \ \alpha_n^{k,q} = e^{1 - r_d^{k,q}}[(s_n^q + m)]_+ \tag{8}$$

Following the setting in original Circle loss, we can deduce the TContrast loss of node $i$ as:

$$l_i^{TC} = \log[1 + \sum_k \sum_q \exp(\gamma(\alpha_n^{k,q}(s_n^q - m) - \alpha_p^{k,q}(s_p^k - (1 - m))))] \tag{9}$$

Since the decision boundary is achieved at $\alpha_n(s_n - m) - \alpha_p(s_p - (1 - m)) = 0$, combined with (7), the decision boundary in TContrast loss is given by:

$$\frac{s_n}{e^{r_d}} + \frac{(s_p - 1)^2}{e^{1 - r_d}} = \frac{m^2}{e^{r_d}} + \frac{m^2}{e^{1 - r_d}} \tag{10}$$

It shows that the decision boundary of TContrast loss is an ellipse arc, where the center is $(0, 1)$, and the lengths of the semi-axis along $s_n$ and $s_p$ are $m \cdot \sqrt{1 + e^{2r_d - 1}}$ and $m \cdot \sqrt{1 + e^{1 - 2r_d}}$. So TContrast loss expects $s_n < m \cdot \sqrt{1 + e^{2r_d - 1}}$ and $s_p > 1 - m \cdot \sqrt{1 + e^{1 - 2r_d}}$. The intuitive interpretation is that we set different relaxations to $s_n$ and $s_p$ according to $r_d$. If $r_d$ increases, it will have a more strict margin for $s_p$ and a looser margin for $s_n$, which means we emphasize on improving $s_p$ when the negative nodes close to target.

In TContrast loss $\mathcal{L}_{TC}$, We formulate different penalty strengths according to the similarity score and relative distances of node pairs. We serve it as a regularizer in our final objective loss function, which also includes L1-loss $\mathcal{L}_{l1}$ of true timing and predicted timing:

$$\mathcal{L}_{final} = \mathcal{L}_{l1} + \mathcal{L}_{TC} = \mathcal{L}_{l1} + \lambda \frac{1}{B} \sum_{i=1}^B l_i^{TC} \tag{11}$$

## 5 Experimental Setting

In this section, we first explain the datasets, parameter settings, and baselines of our experiments, then demonstrate the advantages of our method in the IC timing prediction through the experiments.

**Prediction Task.** Given the RC trees extracted from IC interconnections and the driving voltage at the input node, we aim to predict the pin-to-pin timing including delays and slews. We perform the prediction tasks on two benchmark datasets, which are summarized in Table 1, to comprehensively evaluate the performance of our SyncTREE model. Given SPICE timing results as golden, we use

MAE as the evaluation metric for both delay and slew predictions. All experiments in this paper are implemented with PyTorch 1.13.1 and PyTorch Geometric 2.2.0 frameworks and executed on a Ubuntu server equipped with Intel Xeon Gold 6330 CPU with 56 cores/2 threads running at 2.0GHz. The reference SPICE simulations are carried out with the commercial Synopsys HSPICE simulator on an AMD Ryzen 3950X with 16 cores/32 threads at 3.5GHz. (Code and datasets are available at https://github.com/xlab-ub/SyncTree).

Table 1: Statistics of Datasets

|  | Synthetic Dataset | RISC-V Dataset | | |
| --- | --- | --- | --- | --- |
| Configuration | uniform distribution | min | average | max |
| Nodes | 2-51 nodes | 2 nodes | 6.21 nodes | 20 nodes |
| Resistance | 10-2000 ohm | 0.5 ohm | 101.6 ohm | 549.1 ohm |
| Capacitance | 0.01-2 fF | 0.00113 fF | 0.192 fF | 2.398 fF |
| Samples | 4,066 circuits | 414,639 circuits | | |

**Synthesized Dataset.** The dataset consists of synthesized RC circuits with various circuit typologies, including rare interconnections. To generate these circuits, we first generate an RC tree and then simulate its corresponding artificial IC interconnects with SPICE to obtain the golden timing values. The pseudocode for generating RC trees, the workflow for data preparation, and the golden timing distribution in data samples are provided in the Appendix.

**RISC-V dataset.** The dataset is composed of real RC circuits extracted from practical RISC-V IC designs. Our objective is to validate the effectiveness of our model in analyzing the timing of interconnects in practical IC designs.

**Models and parameters.** For our SyncTREE model, we separately set 32 hidden dimensions on the synthetic dataset and 128 hidden dimensions on the RISC-V dataset. The TContrast loss is set with $hops$ of 2, $N$ of 1, $m$ in the range of $[0.1, 0.5]$, and $B$ is 64 for the synthetic dataset and 128 for the RISC-V dataset. All the models are trained with the Adam optimizer $(\beta_1 = 0.9, \beta_2 = 0.99)$. The batch size is set to 32 for the synthetic dataset and 256 for RISC-V dataset. The learning rate was set to 8e-4 for 4/8/16-layer models and 4e-4/2e-4 for 32/64-layer models. All the models are trained for 10 epochs on the RISC-V dataset and 60 epochs on the synthetic dataset.

**Baselines.** Considering timing analysis is a unique task that depends on the structures of RC trees at different levels, we compare our method to various conventional GNNs, including GCN [29], GAT [27], and GraphSAGE [30], as well as prospective GNNs such as DeepGCN[8], GraphTrans[9], and NeuralTree[10]. For GraphTrans, to incorporate complex global information into node features, we concatenate the CLS token embedding of Transformer with node embeddings and input to MLP to get the final output.

## 6 Results and Discussion

In this section, we present a comprehensive evaluation of our SyncTree model's performance compared to selected GNN baselines for delay and slew prediction tasks and showcase the impact of using TContrast loss function for training. Then, we discuss the computational efficiency of SyncTREE and its transfer learning capabilities.

**Main Results.** We present the delay and slew prediction results of different GNN models on the Synthetic and RISC-V datasets in Table 2 and Table 3. The results show that SyncTREE outperforms the existing GNN baselines across all model depths on both synthetic and RISC-V datasets. For example, even the only four-layer SyncTREE model nearly surpasses all the other baselines with different model depths. Furthermore, our model consistently outperforms the baselines on the RISC-V dataset, highlighting its applicability to real-world scenarios. To evaluate the effectiveness of graph representations obtained by different GNN models, we visualize the final node embeddings of the converged GNNs in Figure 5 and calculate the Pearson correlation coefficient with the target delay distribution across the entire circuit graph. The results show that our model can achieve the highest correlation score compared to other baselines, further validating its superiority.

Table 2: Mean Average Error of Prediction Results in Synthetic Dataset

| Layers | Delay Prediction Error (ps) | | | | | | | | |
|---|---|---|---|---|---|---|---|---|---|
| | Baselines | | | | | | Ours | | |
| | GCN | GAT | GraphSAGE | DeepGCN | GraphTrans | NTREE | SyncTREE | SyncTREE+TC | SyncTREE+C |
| 4 | 12.675 | 11.192 | 9.855 | 9.231 | 9.077 | 9.457 | **8.745** | **8.622** | **8.699** |
| 8 | 13.037 | 9.834 | 9.798 | 9.193 | 9.283 | 9.432 | **6.631** | **6.521** | **6.590** |
| 16 | 13.338 | 7.791 | 9.732 | 9.438 | 7.853 | 8.461 | **3.775** | **3.648** | **3.912** |
| 32 | 13.353 | 6.942 | 10.278 | 9.363 | 9.518 | 8.217 | **3.424** | **3.354** | **3.370** |
| 64 | 13.782 | 13.356 | 10.232 | 9.178 | 8.859 | 7.022 | **3.556** | **3.451** | **3.523** |
| | Slew Prediction Error (ps) | | | | | | | | |
| | Baselines | | | | | | Ours | | |
| | GCN | GAT | GraphSAGE | DeepGCN | GraphTrans | NTREE | SyncTREE | SyncTREE+TC | SyncTREE+C |
| 4 | 37.548 | 33.186 | 32.705 | 29.561 | 29.121 | 29.781 | **25.672** | **25.579** | **25.601** |
| 8 | 39.004 | 35.164 | 33.429 | 29.338 | 29.483 | 29.772 | **20.540** | **20.215** | **20.620** |
| 16 | 40.348 | 27.817 | 34.908 | 29.691 | 28.352 | 28.368 | **14.305** | **14.047** | **14.285** |
| 32 | 41.827 | 25.847 | 34.857 | 29.563 | 27.446 | 28.581 | **12.864** | **12.608** | **13.107** |
| 64 | 43.913 | 42.092 | 36.962 | 29.758 | 28.897 | 27.044 | **14.544** | **14.426** | **14.591** |

Table 3: Mean Average Error of Prediction Results in RISC-V dataset

| Layers | Delay Prediction Error (ps) | | | | | | | | |
|---|---|---|---|---|---|---|---|---|---|
| | Baselines | | | | | | Ours | | |
| | GCN | GAT | GraphSAGE | DeepGCN | GraphTrans | NTREE | SyncTREE | SyncTREE+TC | SyncTREE+C |
| 4 | 0.0467 | 0.0424 | 0.0417 | 0.0430 | 0.0359 | 0.0422 | **0.0313** | **0.0306** | **0.0317** |
| 8 | 0.0403 | 0.0403 | 0.0395 | 0.0372 | 0.0341 | 0.0298 | **0.0195** | **0.0182** | **0.0223** |
| 16 | 0.0343 | 0.0285 | 0.0371 | 0.0453 | 0.0319 | 0.0263 | **0.0128** | **0.0114** | **0.0131** |
| 32 | 0.0321 | 0.0304 | 0.0405 | 0.0311 | 0.0338 | 0.0271 | **0.0106** | **0.0098** | **0.0109** |
| 64 | 0.0379 | 0.0389 | 0.0436 | 0.0450 | 0.0357 | 0.0314 | **0.0176** | **0.0159** | **0.0181** |
| | Slew Prediction Error (ps) | | | | | | | | |
| | Baselines | | | | | | Ours | | |
| | GCN | GAT | GraphSAGE | DeepGCN | GraphTrans | NTREE | SyncTREE | SyncTREE+TC | SyncTREE+C |
| 4 | 0.5453 | 0.4739 | 0.5319 | 0.1514 | 0.1312 | 0.2231 | **0.0410** | **0.0406** | **0.0418** |
| 8 | 0.3719 | 0.3740 | 0.4726 | 0.1864 | 0.1436 | 0.1974 | **0.0265** | **0.0237** | **0.0258** |
| 16 | 0.3141 | 0.3257 | 0.3315 | 0.1908 | 0.1377 | 0.2085 | **0.0252** | **0.0228** | **0.0249** |
| 32 | 0.3004 | 0.2989 | 0.3569 | 0.1190 | 0.1349 | 0.1859 | **0.0464** | **0.0428** | **0.0491** |
| 64 | 0.2944 | 0.3101 | 0.4021 | 0.1875 | 0.1510 | 0.1922 | **0.0506** | **0.0501** | **0.0533** |

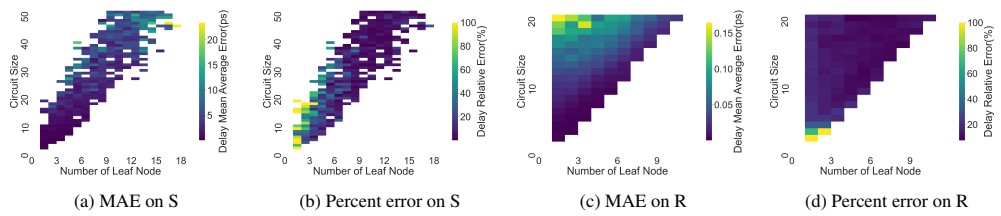

(a) MAE on S     (b) Percent error on S     (c) MAE on R     (d) Percent error on R

Figure 3: Delay Prediction Result of SyncTREE on Synthetic dataset (S) and RISC-V dataset (R).

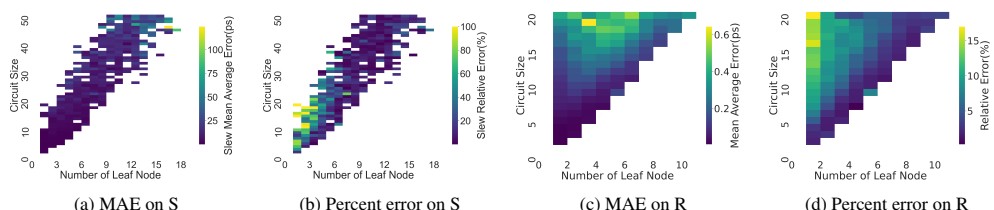

(a) MAE on S     (b) Percent error on S     (c) MAE on R     (d) Percent error on R

Figure 4: Slew Prediction Result of SyncTREE on Synthetic dataset (S) and RISC-V dataset (R).

**Accuracy vs. circuit size and the number of sinks.** To provide more insights into the performance of our method, we investigate the relationship between prediction accuracy and two important factors: circuit size and the number of sinks (i.e., leaf nodes) in RC trees. As depicted in Figure 3 and Figure 4, our method achieves better accuracy on circuits with larger sizes and more sinks.

**TContrast loss vs. Circle loss.** We provide the prediction results of SyncTREE models which trained with TContrast (TC) loss and Circle (C) loss in Table 2 and Table 3, we found that Sync-TREE+TC performs better than the default L1 loss and Circle loss, highlighting the advantages of TC in optimizing similarity pairs and loss optimization convergence. To further compare the original

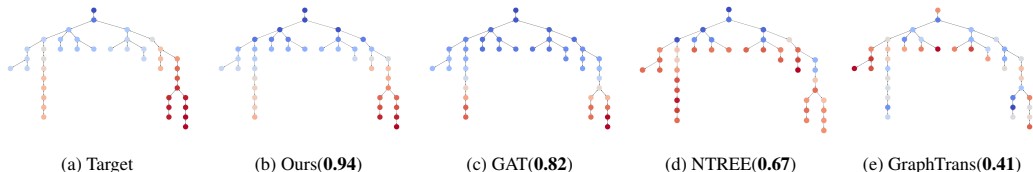

(a) Target       (b) Ours(**0.94**)       (c) GAT(**0.82**)       (d) NTREE(**0.67**)       (e) GraphTrans(**0.41**)

Figure 5: Visualization of final feature embeddings obtained by different GNNs.

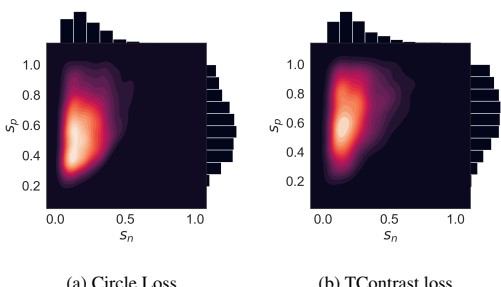

(a) Circle Loss       (b) TContrast loss

Figure 6: Visualization of the similarity distributions after convergence.

Circle loss and TContrast loss, we analyzed their convergence status in the delay prediction task. Figure 6 shows the distribution of similarity pairs $(s_n, s_p)$ after convergence. We observed that the similarity pairs using TContrast loss are closer to the optimization goal and have a more concentrated distribution in similarity space.

**Computational Efficiency.** SPICE typically requires significant computational resources to model circuit behavior in a time-incremental manner. However, considering that RC circuits are linear systems, we can utilize linear multi-step numerical integration techniques like the Trapezoidal method [31] to solve the differential-algebraic equations if the form of $\frac{dx}{dt} = Ax$ inside SPICE for transient simulation. This results would be $x(t + h) = (I - \frac{1}{2}hA)^{-1}(I + \frac{1}{2}hA)x(t)$, given a circuit with $n$ nodes and a driving signal with $m$ time steps from the source. With LU factorization being used to invert $n \times n$ matrices, the computation complexity of SPICE is expressed as $O(m \cdot (n^3 + n^2 + n))$.

In contrast, the computational complexity of our SyncTREE model is significantly reduced, which makes it highly efficient. As it's GAT-based, there are no extensive matrix operations required, and the computations can be parallelized easily. Moreover, the number of layers $l$ is less than $n$. Thus, the time complexity of our model can be expressed as $O(l \cdot (n \cdot f \cdot f' + (n-1) \cdot f'))$, where $f$ is the number of input features, $f'$ is the number of output features, and $n - 1$ is the number of edges. We provide the computational efficiency comparison of SyncTREE with SPICE simulation in Figure 7. Our results indicate that SyncTREE is significantly faster than SPICE in obtaining accurate results, with the advantage increasing as the circuit size grows.

**Transfer Learning.** We provide the results of ablation study of transfer learning of SyncTREE on both delay and slew tasks in Table 4. During transfer learning, we froze the weights pre-trained on the other task and only fine-tuned the readout module on the target task. The results show that the representations learned from the delay prediction task can be efficiently transferred to the slew task, resulting in better performance. The indicates that SyncTREE's learned representation can be efficiently transferred between tasks, which can save time and resources by avoiding the need for additional training on every new target task from scratch.

## 7 Conclusion

In this paper, we propose a novel GNN model to predict circuit timing with a much faster computation speed than the SPICE simulator. Furthermore, based on the RC tree's structural and physical properties, we devise a tree contrastive (TContrast) loss to guide the feature transformation during graph learning. The results show that our model reaches state-of-the-art performance compared

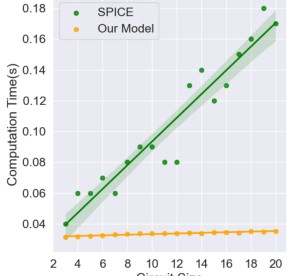

Figure 7: Runtime of SPICE simulation and SyncTREE model with circuit size.

| Layers | MAE Change with Transfer Learning | | | |
| | Delay to Slew | | Slew to Delay | |
| | Synthetic | RISC-V | Synthetic | RISC-V |
|---|---|---|---|---|
| 4 | 0.917↓ | 0.0009↓ | 0.720↑ | 0.0007↑ |
| 8 | 1.598↓ | 0.0021↓ | 0.326↑ | 0.0012↑ |
| 16 | 2.304↓ | 0.0032↓ | 0.343↑ | 0.0026↑ |
| 32 | 1.876↓ | 0.0054↑ | 0.969↑ | 0.0018↑ |
| 64 | 2.068↓ | 0.0037↑ | 0.505↑ | 0.0034↑ |

Table 4: SyncTREE Transfer Experiments Between Delay Prediction and Slew Prediction. (↓ means "lower is better")

with prospective GNNs and outperforms the SPICE simulator in computation efficiency. Therefore, our SyncTREE model can better support fast incremental timing updates. Compared to SPICE's full-fledged simulation even for incremental circuit updates, our solution's incremental updates will be even more appealing to IC design optimization.

## Acknowledgment

This work was supported, in part, by the SUNY-IBM AI Collaborative Research Alliance Project #22003, the SUNY Empire Innovation Program, and, by Advantest as part of the Graduate School "Intelligent Methods for Test and Reliability" (GS-IMTR) at the University of Stuttgart.

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

# A Appendix

### A.1 Message Passing in SyncTREE

As the message passing process in Figure 8, information from leaves, sub-branches, and the whole global structure is first collected following the bottom-up propagation by $GAT_{bu}$. Then, the final node representations of $GAT_{bu}$ are applied to each layer of $GAT_{td}$ to jointly update the node attributes of the corresponding top-down tree with the node embeddings of its previous layer. As the example shown in Figure 8, by designing this two-pass message-passing mechanism, the node features will incorporate the information from different levels and become more expressive. Furthermore, in the top-down tree, the root node can only be updated with synchronized $h_{bu}^L$ since it doesn't have any incoming connection, it ensures that information injection at the source of the top-down tree is fixed which can help to maintain differentiable feature embeddings without over-smoothing.

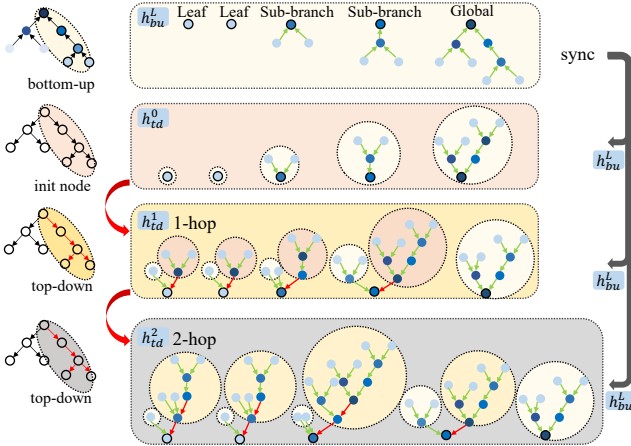

Figure 8: Illustration of our two-pass message-passing mechanism.

### A.2 Synthetic and RISC-V Dataset Preparation

Our dataset is composed of artificially generated and practical RC trees and the golden timing results at sinks (leaf nodes of each RC tree) obtained by SPICE simulation. On the one hand, we follow the pipeline in Figure 9 to generate the synthetic dataset. To be specific, we first adopt Algorithm 1 to generate RC-trees with random typologies and then convert them to artificial IC interconnects for further SPICE timing measurement. On the other hand, we directly extract RC trees from practical RISC-V circuit designs to compose the RISC-V dataset.

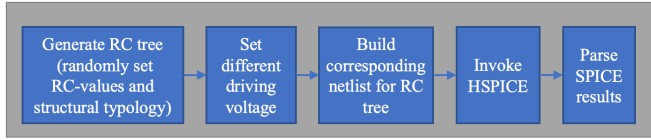

Figure 9: Pipeline for the synthetic dataset generation.

### A.3 Baselines's Implementation

The GNNs of all the baseline models are set with 32/128 hidden dimensions separately for the synthetic dataset/RISC-V dataset. For GraphTrans, the dimension of the feedforward full-connection layers in the Transformer of GraphTrans is set to 256 with 0.1 dropout probability between layers, the number of attention heads is set to 4, and the max input sequence length is set to the maximum circuit size. It should be noted that we only made a little modification to the GraphTrans model. GraphTrans

**Algorithm 1** Generate artificial RC-trees

---

**Require:** $v_d \in [v_{min}, v_{max}], R \in [R_{min}, R_{max}], C \in [C_{min}, C_{max}]$

   Initialize voltage $v_d$ of driving cell, edge type (rising or falling), depth $D$ of RC tree

   $parent\ set = list[drivingcell]$

   $parent \leftarrow$ randomly pick one element from $parent\ set$

   **while** $depth \leq D$ **do**

      randomly choose R, C

      generate $child$, add $child$ into $parent\ set$

      the $R_{child}$ of the edge from $child$ to the $parent \leftarrow R$

      the $C_{child}$ of $child$ to the ground $\leftarrow C$

      $parent \leftarrow$ randomly pick one element from $parent\ set$

      $D = D + 1$

   **end while**

---

is originally designed for node classification tasks, it takes CLS token from Transformer output as the representation of the whole graph and applies a linear module followed by softmax to implement prediction. In order to incorporate global information into node features, in our experiments, we concatenate the CLS token with node embeddings and then feed it into MLP to get the final output. For NTREE, we set GAT as its basic block with a 0.2 dropout probability between layers. We follow the original junction-tree-based algorithm in [10] to compose H-trees from our RC circuits with the same radius setting for extracting subgraphs in the paper.

**A.4 Analysis of TContrast Loss** To visualize the converging process during training, we plot the distribution of similarity pairs in space at different epochs in Figure 10. It obviously shows that our model approaches the optimization goal with a more concentrated similarity distribution after enough training with the guidance of TContrast loss. In Figure 11, we show the MAE difference of timing results obtained by vanilla SyncTREE and TC-loss guided SyncTREE. As shown in the results, after being combined with TC loss, our SyncTREE model has smaller errors for most types of RC trees which can effectively prove the validity of TC loss.

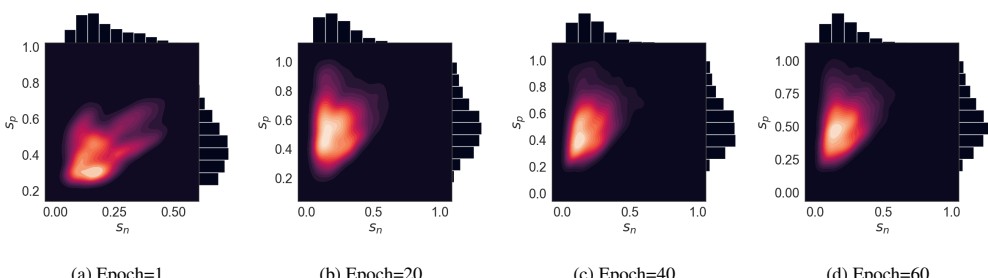

    (a) Epoch=1         (b) Epoch=20         (c) Epoch=40         (d) Epoch=60

Figure 10: The distribution of similarity pairs with training epochs.

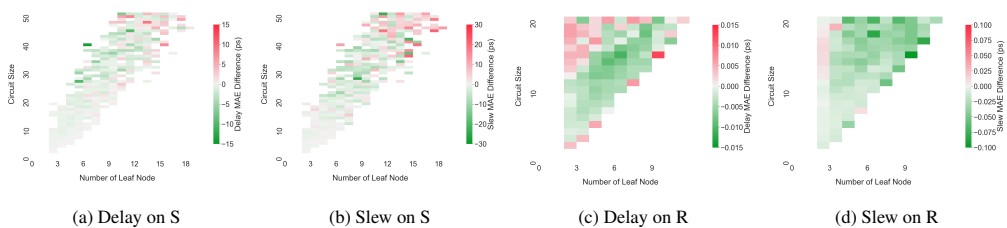

    (a) Delay on S        (b) Slew on S        (c) Delay on R        (d) Slew on R

Figure 11: Mean Average Error difference after applying TContrast loss on the synthetic dataset (S) and RISC-V dataset (R). (Negative values indicate that TC-loss-guided SyncTREE has a lower MAE error than vanilla SyncTREE)

