# OpenReview forum: "SyncTREE: Fast Timing Analysis for Integrated Circuit Design through a Physics-informed Tree-based Graph Neural Network"
_NeurIPS.cc/2023/Conference — NeurIPS 2023 poster_

### Official Review · Reviewer_YDvu · 2023-06-27

**Soundness:** 3 good
**Presentation:** 3 good
**Contribution:** 3 good
**Rating:** 6
**Confidence:** 4

**Summary:**

This papers proposes SyncTREE that uses a bottom-up and top-down graph attention network with a Tree Contrastive Loss to predict the delay and slew for IC interconnects. Compared to other GNN methods, the proposed one achieve lower prediction error across synthetic and RISC-V benchmarks.

**Strengths:**

1.	The proposed method is quite novel and leverages the prior knowledge in timing analysis in network design and loss function design, achieving the best results on both synthetic and RISV benchmarks.

**Weaknesses:**

1.	The modified aggregation mechanism claims to preserve resistance information by linearly combining node and edge features. However, no ablation study or evidence on this technique has been shown. Similarly, the network designs, e.g., residual connections from bottom-up tree, two directed graphs, or other introduced techniques are not evaluated through ablation studies. It seems that the benefits are mainly from the directed graphs, not GAT. How does this method perform if applied to other GNNs listed in Table 3?
2.	The layers in the trained GAT is fixed, not adaptive to different circuits. If a RC tree is deeper than the GAT network, then the timing information cannot be propagated from source to sink. The generalization raises some concerns in this case.
3.	In line 266, it claims the proposed method achieves better accuracy on larger circuits. However, Figure 4(d) seems the relative prediction error get worse with larger circuit sizes if I understand correctly. Why the performance gets worse as circuit size changes is not explained. How to solve this generalization gap should be a key question to investigate for this work.
4.	The runtime comparison need to clarify the hardware platform. If the GAT uses GPU, then it might be a unfair comparison if compared to CPU-version SPICE. There exists GPU version LU factorization to accelerate matrix inversion.
5.	The prediction error lacks intuitive analysis. Whether 0.05 ps MAE considered to be large or small enough for timing analysis is not clear. In other words, the significance of the results is unclear. How does this delay and slew error ultimately affect critical path delay or total negative slack should be discussed. The variance in the predicted error on each node is also important compared to MAE.
6.	In terms of generalization and transfer learning, transferring to different benchmark/circuit are expected instead of transfer from delay to slew prediction. How does the physics-informed loss function helps to increase data efficiency compared to pure data-drive method is not shown. The generalization and data efficiency are critical concerns for ML-based PDE solving tasks, which are not deeply explored in this paper.


**Questions:**

Major questions are summarized above. Basically, more ablation studies are required. More discussion on generalization to different circuits and data efficiency is expected. I will consider increasing the score if the major concerns above are addressed.

**Limitations:**

No.

---

> ### Author Rebuttal · Authors · 2023-08-10
>
> ## Response to Reviewer #YDvu
> Thank you for carefully going through the paper and pointing out so valuable questions. We hope these responses satisfactorily answer your questions:
>
> ### Q1. Ablation study.
> 1. Ablation study toward GAT modification.
>
> Following the reviewer’s suggestion, we evaluate our model with/without modification towards the GAT aggregation mechanism (GAT-Mod). The results are as follows:
>
> Table 1 MAE ps on Synthetic Dataset w/wo GAT-Mod
> |             | 4 Layers | 8 Layers | 16 Layers | 32 Layers | 64 Layers |
> |-------------|---------:|---------:|----------:|----------:|----------:|
> | W-GAT-Mod   |   **8.745**   |   **6.631**   |    **3.775**    |    **3.424**    |    **3.556**    |
> | WO-GAT-Mod  |    9.752    |    6.589    |     4.886     |     4.597     |     4.507     |
>
> Table 2 MAE ps on RISC-V Dataset w/wo GAT-Mod
>   |            | 4 Layers | 8 Layers | 16 Layers | 32 Layers | 64 Layers |
>   |------------|----------|----------|-----------|-----------|-----------|
>   | W-GAT-Mod  | **0.0313** | **0.0195** | **0.0128** | **0.0106** | **0.0176** |
>   | WO-GAT-Mod | 0.0395   | 0.0325   | 0.0274    | 0.0271    | 0.0294    |
>
> 2. Other discussions.
> - For the residual connections from the bottom-up tree, if we remove this part, the model will degrade to conventional GAT working on a directed graph. In this case, the leaf nodes only can gather information from the propagation path, leading to the loss of high-level substructure features.
> - About applying two directed graphs to other GNNs, let us take the results in Table 2 and 3 as reference. Among all baseline models, GAT presents higher accuracy. That's why we use GAT as the basic block in our SyncTREE model.
>
> ### Q2. Concern about the depth of RC trees and the number of convolution layers.
> It’s true that if an RC tree is deeper than the GAT network, then the timing information cannot be propagated from source to sink. This is a typical issue of message-passing GNNs caused by its inner mechanism. A deeper graph structure might necessitate a deeper GNN architecture to capture long-range dependencies. To show the benefit of adding layers to deeper RC trees and the accuracy changes it brings to shallower circuits, we analyze the relative error regarding RC tree depth and size under different model depths. We attach this result in the pdf of the global response, please refer to Fig.1 and Fig. 2.
>
> ### Q3.1) Clarification about Fig 4d’s description; 2) Generalization problem.
> 1) We are sorry for our unclear description regarding Fig. 4d’s description. The results in Fig. 3 and Fig. 4 show that our approach demonstrates enhanced accuracy when applied to circuits of greater dimensions and more sinks.
>
> 2) To evaluate our model's performance thoroughly, we make two highly diverse datasets, as shown in statistics shown in Fig. 10 and Fig. 11 in our supplementary material. Our model is trained on datasets containing various RC circuits of different sizes and typologies. Compared with golden delay, the value range of golden slews is more limited, which makes the model tend to predict the small slew values very well as shown in Fig. 4d. However, delays are highly varied across circuit sizes, for example, delays of tiny circuits (eg. RC trees that have less than 4 nodes) are only at 1e-4 ps level, leading to higher errors for these circuits as shown in Fig. 3d. It should be pointed out that these tiny circuits are not common in normal IC designs but deserve to be investigated with our model. As for the results in Fig. 4d showing increasing error with larger circuit sizes, we believe the overall accuracy is still quite satisfactory considering the high diversity of our dataset.
>
> ## Q4. Running Platform Details.
> We report this question in the global response.
>
> ## Q5. 1) Interpretation towards MAE value; 2) Significance of our work; 3) Error impact on Critical path delay.
> We report problems 1 and 2  in the global response.
>
> 3) We follow the reviewer’s suggestion and calculate the critical delay MAE on both two datasets. The critical delay MAE is the average of absolute error between the golden critical delays and the predictions at the critical sinks across all circuits in the dataset. Please note the average critical delays on Synthetic dataset and RISC-V dataset are 183.25 ps and 0.68 ps respectively.
>
> |                         |  4L   |  8L   | 16L   | 32L   | 64L   |
> |-------------------------|-------|-------|-------|-------|-------|
> | Synthetic Critical-MAE  | 78.828| 59.976| 32.004| **24.837** | 28.661|
> | RISC Critical-MAE       | 0.4034| 0.2170| 0.1311| **0.1214** | 0.1283|
>
> From the result, we can observe that the critical delay MAE is closely related to overall MAE.
>
> ## Q6. 1) Transfer learning to different benchmarks/circuits; 2) The advantage over data efficiency of Physics-informed loss function.
> 1) Our model is trained on datasets containing various RC circuits of different sizes and typologies, which means that our model doesn’t need to be trained differently regarding different circuit sizes. Besides, our two benchmark datasets are from different sources, thus we have different hidden dimension settings to optimize the accuracy correspondingly, so it's not feasible do transfer learning to different benchmarks due to the model difference and dataset inconsistency.
>
> 2)  In terms of data efficiency, we implement an additional experiment to evaluate the possible benefit of TC loss under different training set percentages.
>
> |        | Synthetic-W-TC | Synthetic-WO-TC | RISC-V-W-TC | RISC-V-WO-TC |
> |--------|---------------|----------------|-------------|--------------|
> | 25%    | 4.2838        | 4.5682         | 0.0311      | 0.0342       |
> | 50%    | 4.0534        | 4.0246         | 0.0166      | 0.0228       |
> | 75%    | 3.9063        | 3.8453         | 0.0149      | 0.1912      |
>
> From the result, we can observe that SyncTREE with TC loss exhibits obvious benefits over data efficiently on the RISC-V dataset.

---

> > ### Comment · Reviewer_YDvu · 2023-08-15
> > **Thanks for the response**
> >
> > Thanks for the response. The authors mostly addressed my questions. The overall prediction error in terms of MAE versus the average critical delay is still not small enough for accurate timing prediction, which needs further improvement. I will increase the score to 6.

---

> > > ### Author Response · Authors · 2023-08-17
> > >
> > > Dear reviewer #YDvu,
> > >
> > > We are pleased that our rebuttal addressed your concerns. We greatly appreciate your recognition of our work!

---

### Official Review · Reviewer_sZvh · 2023-07-05

**Soundness:** 2 fair
**Presentation:** 2 fair
**Contribution:** 3 good
**Rating:** 3
**Confidence:** 4

**Summary:**

This manuscript present a SyncTree to speed up timing analysis in IC design.

**Strengths:**

-The problem that the manuscript is trying to address is important (increase speed of timing analysis)
-Evaluation and comparison to other machine learning based methods.


**Weaknesses:**

- It is unclear are the Mean Average Error with respect to Spice simulation?
- Table 2 and Table 3 should consist of time it takes to perform the predictions.
- Additionally, what is the increase in speed between SyncTree and spice simulation.

**Questions:**

Check Weakness section

**Limitations:**

Check Weakness section

---

> ### Author Rebuttal · Authors · 2023-08-10
>
> ## Response to Reviewer #sZvh
> Thank you for your comments and the time spent reviewing the work. We try to address all the raised points in the following content.
> ### Q1. Clarification about MAE.
> In our experiments, we treat the SPICE simulation measurements as the golden timing results. The Mean Average Error (MAE) is measured between predictions and spice simulation results.
> ### Q2. Running time of results in Table 2 and Table 3.
> We additionally record the inference time (s) of results in Table 2 and Table 3 as follows:
> 1. Table 2
> |    | GCN    | GAT    | GraphSAGE | DeepGCN | GraphTrans | NTREE | SyncTREE |
> |----|--------|--------|-----------|---------|------------|-------|----------|
> | 4L | 0.003  | 0.005  | 0.002     | 0.008   | 0.060      | 0.024 | 0.015    |
> | 8L | 0.005  | 0.009  | 0.004     | 0.015   | 0.055      | 0.038 | 0.025    |
> | 16L| 0.011  | 0.020  | 0.007     | 0.029   | 0.059      | 0.072 | 0.044    |
> | 32L| 0.019  | 0.039  | 0.013     | 0.055   | 0.062      | 0.114 | 0.100    |
> | 64L| 0.036  | 0.071  | 0.025     | 0.112   | 0.056      | 0.233 | 0.185    |
>
> 2. Table 3
> |     | GCN    | GAT    | GraphSAGE | DeepGCN | GraphTrans | NTREE | SyncTREE |
> | --- | ------ | ------ | --------- | ------- | ---------- | ----- | -------- |
> | 4L  | 0.007  | 0.008  | 0.006     | 0.018   | 0.101      | 0.057 | 0.035    |
> | 8L  | 0.010  | 0.015  | 0.008     | 0.033   | 0.099      | 0.110 | 0.051    |
> | 16L | 0.017  | 0.031  | 0.015     | 0.070   | 0.085      | 0.193 | 0.097    |
> | 32L | 0.033  | 0.060  | 0.029     | 0.131   | 0.084      | 0.245 | 0.162    |
> | 64L | 0.0659 | 0.1172 | 0.0665    | 0.269   | 0.087      | 0.317 | 0.294    |
>
> ### Q3.  Improvement in speed of SyncTREE.
> As shown in Fig. 7, we evaluate the computation efficiency of SyncTree model and SPICE simulation along with the circuit size. It shows that SyncTREE is significantly faster than SPICE with this advantage increasing as the circuit size grows. We further analyze the computation complexity towards SPICE and SyncTREE in the Computation Efficiency Section of Results and Discussion Part. To conclude, SPICE needs to solve DAEs in a time-incremental manner to simulate the circuit behavior, which involves intensive matrix operations like LU decomposition. In contrast, SyncTREE leverages Graph Attention Networks which only revolve around linear transformation, leading to a much faster running speed.

---

> > ### Author Response · Authors · 2023-08-21
> >
> > Dear Reviewer #sZvh,
> >
> > Thank you so much for your time and efforts in reviewing our paper. If any sections of our rebuttal you felt unclear, could you kindly point out them? We will appreciate your invaluable insights.
> >
> > Thank you again. Looking forward to your continued guidance.

---

> ### Comment · Area_Chair_yKxC · 2023-08-16
> **Respond to authors' rebuttal**
>
> Please, look at the authors' rebuttal and the other reviewers' comments and indicate if you would like to change anything in your review.

---

> > ### Comment · Area_Chair_yKxC · 2023-08-19
> > **Reminder**
> >
> > A reminder of this.

---

### Official Review · Reviewer_6pFT · 2023-07-07

**Soundness:** 3 good
**Presentation:** 3 good
**Contribution:** 3 good
**Rating:** 6
**Confidence:** 1

**Summary:**

This paper proposes a GNN based method that specializes in timing analysis.

**Strengths:**

* This paper is well written and organized, easy for readers to follow.
* Related background, related work that uses GNN on circuits are discussed with sufficient level of detail.
* The core problem looks well formatted.
* Experimental results looks promising with benchmarks on RISC-V, offering faster than SPICE simulation looks appealing.
* Performance and results are thoroughly analyzed.


**Weaknesses:**

Please see questions

**Questions:**

* For different circuit technology sizes, does the proposed method need to be trained differently? or is the technology sizing part of the input?


**Limitations:**

* There is no explicit discussion on limitations.

---

> ### Author Rebuttal · Authors · 2023-08-10
>
> ## Response to Reviewer #6pFT
> Thank you for your feedback and support! We hope these responses satisfactorily answer your questions:
> ### Q1. Model adaptation to different circuit size.
> In this paper, our motivation is to devise a timing prediction model that can be applied to different circuits by leveraging the representation capability of Graph Neural Networks for unstructured data. Our model is trained on the dataset containing various RC circuits of different sizes and typologies, which means that our model doesn’t need to be trained differently regarding different circuit sizes.

---

> ### Comment · Area_Chair_yKxC · 2023-08-16
> **Respond to authors' rebuttal**
>
> Please, look at the authors' rebuttal and the other reviewers' comments and indicate if you would like to change anything in your review.

---

> ### Author Response · Authors · 2023-08-21
>
> Dear reviewer #6pFT,
>
> Thank you very much for your recognition of our paper. If any parts of the rebuttal you felt were unclear, could you please kindly highlight them? We will be very appreciative of your insights.

---

### Official Review · Reviewer_ZUNz · 2023-07-30

**Soundness:** 3 good
**Presentation:** 2 fair
**Contribution:** 3 good
**Rating:** 5
**Confidence:** 2

**Summary:**

The paper proposes a GNN model, dubbed SyncTREE, for IC's RC-tree timing analysis. Two techniques are proposed: 1) two-pass message-passing and 2) Tree Contrastive loss. Experiments of two IC designs demonstrate the best accuracy of SyncTREE over other SOTA GNN models.

**Strengths:**

1. Domain-specific knowledge is used in SyncTREE's two techniques. I think these two techniques are original.

2. Evaluations demonstrate the performance of SyncTREE's two techniques. The timing analysis shows the promising potential speedup of SyncTREE over traditional SPICE with increased circuit sizes.

**Weaknesses:**

1. There is no clear guideline on setting the number of hidden dimensions and layer selection, which can influence accuracy and speed trade-offs for the practical adoption of the proposed SyncGNN.

2. I'm pondering if an EDA-focused conference like DAC or ICCAD may be a more suitable platform for this paper. Given that the development of SyncGNN draws heavily on EDA-related domain-specific knowledge, such a conference could provide a more targeted audience and potentially foster more fruitful discussions for further refining this approach.

**Questions:**

1. I am confused by the middle part of Fig. 2 w/ "Init Nodes". The hidden features from the bottom-up graph are used to initialize those in the top-down graph. Why there is an arrow from the top-down graph to the "Init Nodes"? In addition, the right-most part of Fig. 2 shows a "mask", which is given w/o explanation.

2. What is the rule of thumb for setting the number of hidden dimensions? In addition, it would be better if the authors can discuss the selection of the number of layers. These hyperparameters can influence the accuracy and speed trade-offs and a discussion can potentially ease the use of the proposed SyncTREE.

3. Fig.7 provides the computation time vs. circuit size. What are the hardware platforms for SPICE simulation and SyncTREE model? Since given a new circuit, new graph training is needed. It would be better if the author can also provide the training time vs. circuit size for better show the efficiency of the proposed SyncTREE.

4. Typos. E.g., "news ways" should be "new ways" in the Abstract section. Line # 216, "We" should be "we".

5. Considering that timing analysis is an important problem for EDA community and considering existing efforts to use GNN for EDA tasks, can conferences such as DAC and ICCAD be a more suitable platform for publishing this paper?

**Limitations:**

The authors do not discuss the limitations of the proposed method. It would be better if the authors can provide the limitations and the potential future directions of this work.

---

> ### Author Rebuttal · Authors · 2023-08-10
>
> ## Response to Reviewer #ZUNz
>
> We greatly appreciate your careful and detailed review. Here are some points we would like to clarify:
>
> ### Q1. Explanation of “Init Nodes” & “mask” in Fig. 2.
>
> The "Init Nodes" part of Fig. 2 involves the node attributes assignment of the top-down graph after each convolutional layer. In our two-pass message-passing mechanism, as shown in Equation (5), the node attributes in the top-down graph at $l+1$ layer will be updated by two parts: 1) the node representations obtained by the $AGG^{l}_{td}$; 2) the final node embeddings of the bottom-up graph. With this update mechanism, the node attributes will be reassigned accordingly for the next layer.
>
> The mask in Fig. 2 is used to filter out leaf nodes for calculating the loss function and obtaining the final output. Since our task is to predict propagation delay/slew of RC trees, we take the final representations at leaf nodes (sinks) as the prediction results. That's why we apply masks to improve the model performance by reducing the influence of irrelevant information in the loss function.
>
> ### Q2. Rules regarding setting hidden dimensions and the number of convolution layers.
> The hyperparameter selection of hidden dimensions and the number of convolution layers directly determine the model performance. A lower hidden dimension might lead to underfitting, leading to the failure of the model to capture complex patterns in the graph data. Conversely, a higher hidden dimension could lead to overfitting. Besides, the number of convolutional layers in GNNs determines the depth of information propagation across the graph. To choose an ideal number of convolution layers, the depth of input graphs is a crucial factor to consider. In our work, the choice of hidden dimensions and the number of convolution layers is decided by the trade-off between model accuracy and efficiency. By systematically exploring different parameter combinations and analyzing the model’s behavior, we then make an informed decision about the appropriate number of layers and dimensions to apply.
>
> Following the reviewer’s suggestion, we evaluate the inference time and model accuracy on the Synthetic and RISC-V datasets under different parameter combinations. The results are shown in the following tables (Note: 4L refers to 4 convolution layers, 32D refers to 32 hidden dimensions).
>
> Table 1 MAE ps (Inference Time s) on Synthetic Dataset (batches: 31, batch size: 32)
> |        |  4L         |  8L         | 16L         | 32L         | 64L         |
> |--------|------------|------------|-------------|-------------|-------------|
> |  32D   | 8.745 (0.010) | 6.631 (0.018) | 3.775 (0.035) | **3.424** (0.068) | 3.556 (0.141) |
> |  64D   | 9.075 (0.014) | 7.258 (0.026) | 4.886 (0.051) | 4.662 (0.102) | 4.563 (0.204) |
> | 128D   | 9.564 (0.022) | 6.986 (0.041) | 4.737 (0.079) | 4.419 (0.157) | 5.106 (0.314) |
>
>
> Table 2 MAE ps (Inference Time s) on RISC-V Dataset (batches: 647, batch size: 128)
> |       | 4L              | 8L              | 16L             | 32L               | 64L              |
> |-------|-----------------|-----------------|-----------------|-------------------|------------------|
> | 32D   | 0.0385 (0.017)  | 0.0569 (0.021)  | 0.0213 (0.039)  | 0.0145 (0.075) | 0.0244 (0.136)  |
> | 64D   | 0.0352 (0.021)  | 0.0271 (0.034)  | 0.0149 (0.056)  | 0.0120 (0.103) | 0.0236 (0.202)  |
> | 128D  | 0.0313 (0.033)  | 0.0195 (0.054)  | 0.0128 (0.096)  | **0.0106** (0.160) | 0.0176 (0.312)  |
>
> Given the above statistics, we then separately set 32 hidden dimensions and 128 hidden dimensions for our model over Synthetic and RISC-V dataset.
>
> ### Q3. 1) Hardware platforms running SPICE & our experiments; 2) Plotting training time vs. circuit size.
> 1) All experiments in this paper are implemented with PyTorch 1.13.1 and PyTorch Geometric 2.2.0 frameworks, and executed on a Ubuntu server equipped with Intel Xeon Gold 6330 CPU with 56 cores/2 threads running at 2.0GHz. The reference SPICE simulations are carried out with the commercial Synopsys HSPICE simulator on an AMD Ryzen 3950X with 16 cores/32 threads at 3.5GHz.
>
> 2) As shown in Table 1, Fig. 10, and Fig. 11, our training and validation dataset is composed of RC trees with different sizes and typologies. During model training, all circuit samples are shuffled randomly to reduce bias and improve generalization. Technically, a training batch is composed of various circuits of different sizes and typologies, so it's hard to get the exact training time regarding specific circuit sizes. Moreover, since graph neural networks can deal with unstructured data, if given a new circuit, we don't need to retrain the model. Actually, in the inference stage of our model, the input circuit graphs are new and not seen during training.
>
> ### Q4. Typos correction.
> We appreciate your keen attention to pointing out the typos and apologize for any oversight on our part. We will correct the errors you mentioned in the revision.
>
> ### Q5. A more suitable platform for this paper.
>
> We appreciate your thoughtful consideration of the paper's relevance to different conference platforms within the EDA community. However, we believe that NeurIPS remains a suitable platform for the publication of our work for the following reason.
>
> Our work is mainly about physics-informed deep learning, which combines principles from physics in a specific field with deep learning techniques, and aligns well with the interdisciplinary nature of NeurIPS. To give examples, there are some recent EDA-related papers accepted by premier AI conferences, which makes us firmly believe NeurIPS can effectively draw attention from both the machine learning and EDA communities and be more confident to publish our work through this platform.

---

> ### Comment · Area_Chair_yKxC · 2023-08-16
> **Respond to authors' rebuttal**
>
> Please, look at the authors' rebuttal and the other reviewers' comments and indicate if you would like to change anything in your review.

---

> > ### Comment · Area_Chair_yKxC · 2023-08-19
> > **Reminder**
> >
> > A reminder of this.

---

> ### Author Response · Authors · 2023-08-21
>
> Dear reviewer #ZUNz,
>
> Thank you very much for your time and efforts in reviewing our paper. If any parts of the rebuttal or the paper felt unclear or ambiguous, could you kindly highlight them? Your insights will be greatly appreciated.

---

### Author Rebuttal · Authors · 2023-08-10

## General Response for Common Questions
Thanks for all reviewers' constructive suggestions, which help us find some points we didn’t explain clearly. We believe it’s necessary to make some global clarifications toward the following points:

### 1. Significance of our work.
Timing analysis is crucial for ensuring the proper functioning, performance, and reliability of integrated circuits. Fast and accurate timing prediction can greatly reduce the runtime overhead, which is very significant for IC design. In this paper, we propose a tree-based graph neural network, SyncTREE, to speed up the timing analysis, the significance of our work is listed below:

- To the best of our knowledge, this is the first work applying GNNs to directly make timing predictions, which can offer a good baseline for future works.
- Our model extends beyond conventional GNNs and presents the best representation power for RC trees.
- Compared with SPICE, our model achieves satisfactory accuracy with far less runtime overhead, which means our model has the potential to expedite the IC design process.
- Conventional timing analysis tools like static timing analysis (STA) have limited parallelism capability, slowing down analysis and optimization tasks, while our model is GAT-based and thus can be parallelized easily.

### 2. Dataset clarification.
Our Synthetic dataset and RISC-V dataset are composed of various circuits having different sizes and structures. In our works, we devise a timing prediction model that can be applied to different circuits by leveraging the representation capability of Graph Neural Networks for unstructured data. When given a new and unseen circuit, we don’t need to retain the model.

### 3. Running platform information
All experiments in this paper are implemented with PyTorch 1.13.1 and PyTorch Geometric 2.2.0 frameworks and executed on a Ubuntu server equipped with Intel Xeon Gold 6330 CPU with 56 cores/2 threads running at 2.0GHz. The reference SPICE simulations are carried out with the commercial Synopsys HSPICE simulator on an AMD Ryzen 3950X with 16 cores/32 threads at 3.5GHz.

### 4. MAE metric interpretation
Acceptable propagation delay MAE depends on the specific requirements and performance criteria of the circuit and its intended application. Some applications, such as high-performance computing or communication systems, demand minimizing propagation delay error to ensure accurate timing and reliable operation. Considering the current highest clock rate of CPU doesn’t surpass 10 GHz, which means the time unit for one execution cycle of CPU is above 100 ps. Therefore the 0.05 ps error example reviewer #YDvu posted for IC design is pretty small. Moreover, it should be pointed out that since our goal is to compare with SPICE from both accuracy and computation cost, we set timing results obtained by SPICE simulation as golden and present the percent error between SPICE and our model in Fig. 3 (b) (d) and Fig.4 (b) (d).

---

### Decision · Program_Chairs · 2023-09-21

**Decision:**

Accept (poster)

**Comment:**

Summary:

This paper proposes SyncTREE which uses a bottom-up and top-down graph attention network with a Tree Contrastive Loss to predict the delay and slew for IC interconnects. Compared to other GNN methods, the proposed one achieves lower prediction error across synthetic and RISC-V benchmarks.

Strengths:

1 - Domain-specific knowledge is used in SyncTREE's two techniques which are original.

2 - Evaluations demonstrate the performance of SyncTREE's two techniques. The timing analysis shows the promising potential speedup of SyncTREE over traditional SPICE with increased circuit sizes.

3 - This paper is well-written and organized, easy for readers to follow.

4 - Related background, related work that uses GNN on circuits are discussed with sufficient level of detail.

5 - The core problem looks well formatted.

6 - Experimental results look promising with benchmarks on RISC-V, offering faster than SPICE simulation looks appealing.

7 - Performance and results are thoroughly analyzed.

8 - The problem that the manuscript is trying to address is important (increase speed of timing analysis)

9 - The proposed method is quite novel and leverages the prior knowledge in timing analysis in network design and loss function design, achieving the best results on both synthetic and RISV benchmarks.

Weaknesses:

1 - There is no clear guideline on setting the number of hidden dimensions and layer selection, which can influence accuracy and speed trade-offs for the practical adoption of the proposed SyncGNN.

2 - No ablation study or evidence on this technique has been shown.

3 - The layers in the trained GAT is fixed, not adaptive to different circuits. If a RC tree is deeper than the GAT network, then the timing information cannot be propagated from source to sink. The generalization raises some concerns in this case.

4 - The runtime comparison needs to clarify the hardware platform.

5 - The prediction error lacks intuitive analysis.

Decision:

A majority of reviewers vote for acceptance. The only review favouring rejection is very short and of low quality and the reviewer did not respond to the authors' rebutall or comments by the area chair. This reviewer also does not give a clear reason for rejection. I, therefore, decided to ignore that negative review. Based on this, I decide to accept the paper and encourage the authors to use the feedback provided to improve the paper for its camera ready version.